# Roles of ABCC1 and ABCC4 in Proliferation and Migration of Breast Cancer Cell Lines

**DOI:** 10.3390/ijms21207664

**Published:** 2020-10-16

**Authors:** Floren G. Low, Kiran Shabir, James E. Brown, Roslyn M. Bill, Alice J. Rothnie

**Affiliations:** College of Health & Life Sciences, Aston University, Aston Triangle, Birmingham B4 7ET, UK; low_guy@hotmail.com (F.G.L.); shabirk@aston.ac.uk (K.S.); j.e.p.brown@aston.ac.uk (J.E.B.); R.M.Bill@aston.ac.uk (R.M.B.)

**Keywords:** MRP1, MRP4, breast cancer, proliferation, migration, invasion, cAMP

## Abstract

ABCC1 and ABCC4 utilize energy from ATP hydrolysis to transport many different molecules, including drugs, out of the cell and, as such, have been implicated in causing drug resistance. However recently, because of their ability to transport signaling molecules and inflammatory mediators, it has been proposed that ABCC1 and ABCC4 may play a role in the hallmarks of cancer development and progression, independent of their drug efflux capabilities. Breast cancer is the most common cancer affecting women. In this study, the aim was to investigate whether ABCC1 or ABCC4 play a role in the proliferation or migration of breast cancer cell lines MCF-7 (luminal-type, receptor-positive) and MDA-MB-231 (basal-type, triple-negative). The effects of small molecule inhibitors or siRNA-mediated knockdown of ABCC1 or ABCCC4 were measured. Colony formation assays were used to assess the clonogenic capacity, MTT assays to measure the proliferation, and scratch assays and Transwell assays to monitor the cellular migration. The results showed a role for ABCC1 in cellular proliferation, whilst ABCC4 appeared to be more important for cellular migration. ELISA studies implicated cAMP and/or sphingosine-1-phosphate efflux in the mechanism by which these transporters mediate their effects. However, this needs to be investigated further, as it is key to understand the mechanisms before they can be considered as targets for treatment.

## 1. Introduction

Breast cancer is the most common cancer in women, affecting more than two million women worldwide per year [1]. The development of new targeted therapeutics and the encouragement of women to carry out early screening programs have significantly improved survival rates in the Western world [2,3,4,5]. Breast cancers that express the estrogen receptor and/or the progesterone receptor can be treated with receptor-blocking hormone therapy or with aromatase inhibitors to decrease the levels of estrogen produced [6,7]. Cancers that have high expression of human epidermal growth factor receptor 2 (HER2) can be treated with monoclonal antibodies, which bind to the receptor and block it [7]. However, some breast cancers, termed triple-negative, do not express any of these receptors, and thus, the only treatment available is conventional chemotherapy. Triple-negative breast cancers make up around 10–15% of all breast cancer cases [8], are typically aggressive, and show high levels of metastases and mortality [9,10]. The development of metastasis and/or cancers becoming resistant to therapeutic agents is a growing problem. Women diagnosed at an early stage of breast cancer may have recurrent disease, and at least a quarter of all cases may develop a resistance to therapeutic treatments [2]. Moreover, with the increase in disease progression, the incidence of therapeutic resistance becomes more alarming.

One common cause of therapeutic resistance in breast cancer is the efflux of drugs by membrane proteins of the ATP-binding cassette (ABC) transporter family of proteins [11,12]. ABCC1 (multidrug resistance protein 1/MRP1) and ABCC4 (multidrug resistance protein 4/MRP4) are two members of the C subfamily of ABC transporters that are capable of effluxing several different chemotherapeutic drugs out of cancer cells [13,14,15]. Previous studies have shown that ABCC1 expression is a negative prognostic marker, associated with a decreased survival rate in breast cancer patients [16,17,18] and an increased risk of relapse [19]. Following chemotherapeutic treatment, the expression of ABCC1 in breast cancer tumors was found to increase [20], and its expression level was shown to be highest in the most aggressive subtypes of breast cancers [18]. ABCC4 expression is similarly upregulated in chemotherapy-treated breast tumors compared to noncancerous tissue [21], and ABCC4 polymorphisms are linked with the response to aromatase inhibitors for estrogen-receptor-positive breast cancer [22].

However, it has been proposed that ABCC1 and ABCC4 may play a role in the hallmarks of cancer development and progression, independent of their drug efflux capabilities [23]. This is because ABCC1 and ABCC4 are capable of transporting numerous physiological substrates and have roles in metabolism and inflammation [13,24,25]. For example, ABCC1 can transport glutathione and inflammatory mediators such as leukotrienes and prostaglandins, as well as the bioactive lipids sphingosine-1-phosphate (S1P) and lysophosphatidyl inositol (LPI), which are implicated in cell proliferation, migration, and invasion [26,27,28,29,30,31]. ABCC4 can efflux the cyclic nucleotides cAMP and cGMP involved in cellular signaling, as well as leukotrienes, prostaglandins, and thromboxanes and S1P [32,33,34,35,36,37,38]. Studies on neuroblastoma (a rare childhood cancer) have confirmed that both ABCC1 and ABCC4 play an important physiological role in its development, independent of their role in multidrug resistance, affecting cellular proliferation, migration, and differentiation [39]. ABCC4 has also been implicated in cancer cell proliferation in leukemia [40,41], gastric cancer [42], lung cancer [43], renal cancer [44], ovarian cancer [45], and pancreatic cancer [46,47]. However, less is known about whether ABCC1 and ABCC4 have a role in breast cancer development and/or progression.

In this study, the role of ABCC1 and ABCC4 in breast cancer progression was investigated. The breast cancer cell lines MDA-MB-231 and MCF-7 were used. Both are considered to be aggressive, but MCF-7 is a luminal-type breast cancer with the presence of progesterone, estrogen, and human epidermal growth factor 2 (HER2) receptors, whereas MDA-MB-231 is a basal-type triple-negative cell line. The effect of small molecules inhibitors or siRNA knockdown of ABCC1 and ABCC4 on cellular proliferation, clonogenic capacity, cell migration, and invasion were investigated.

## 2. Results

### 2.1. Expression of ABCC1 and ABCC4 in Breast Cancer Cell Lines

In order to determine whether the breast cancer cell lines expressed ABCC1 or ABCC4, membrane extracts were prepared and assayed via Western blot, as shown in Figure 1. It can be seen that both MCF-7 and MDA-MB-231 cells express both ABCC1 and ABCC4, with the level of expression of the two transporters being higher in the MDA-MB-231 cells than in the MCF-7 cells. In addition, the cell lines were tested for the well-known multidrug resistance transporters P-glycoprotein/ABCB1 and ABCG2 (Appendix A). Only the MCF-7 cell line showed significant expression of these two transporters.

### 2.2. The Effect of ABCC Small Molecule Inhibitors on Breast Cancer Cell Proliferation

Having established that both cell lines expressed the ABCC proteins, we investigated the effect of small molecule inhibitors of these proteins on the ability of the cells to proliferate. The small molecule inhibitors used in the study are detailed in Table 1.

The first parameter investigated was the clonogenic capacity of the cells, i.e., the ability to reproduce from a single cell. Examples of the assay are shown in Figure 2a,b, where it can be seen that, following the treatment of MDA-MB-231 cells with MK571 or Reversan, at increasing concentrations, both the number of colonies formed over seven days and the size of those colonies are reduced. The average data shown in Figure 2c–e show that Reversan and MK571 affect the colony formation for both MDA-MB-231 and MCF-7 cells, but the effect is more pronounced for the MDA-MB-231 cells, perhaps correlating with the higher level of ABCC protein expression in these cells. In contrast, Ceefourin 1 and 2 and Indomethacin had no effect at all on the colony formation.

The effect of these inhibitors on cellular proliferation was also investigated using an MTT assay. As can be seen in Figure 3, the presence of the inhibitors did not affect the proliferation of either cell line for the first 24 h. However, after this time, MK571 and Reversan had a significant impact on the proliferation of both MCF-7 and MDA-MB-231 cells, whereas Ceefourin 1 and 2 and Indomethacin did not. To confirm the results obtained with the MTT assay, and to make sure it was not due to an indirect effect on the enzyme required to reduce MTT, proliferation was also measured by a trypan blue exclusion and cell counting approach (Appendix A). Although the errors are larger using this approach, the key findings replicate those of the MTT assay.

### 2.3. Effect of Inhibitors on Breast Cancer Cell Migration

In addition to rapid proliferation, enhanced migration is a hallmark of aggressive cancers. Therefore, the effects of ABCC inhibitors on breast cancer cell migration was measured using a scratch assay, as shown in Figure 4a. The MDA-MB-231 cells migrated faster than the MCF-7 cells (Figure 4b,c). Most of the inhibitor treatments had no significant effect on the migration. However, the treatment with MK571 did significantly decrease the migration of MDA-MB-231 cells (Figure 4c). This was not due to an effect on proliferation, since after 10–12 h when the migration was most affected, no effect on the proliferation was observed (Figure 3c).

MK571, which inhibits both ABCC1 and ABCC4, and Reversan, which inhibits ABCC1, were the only inhibitors to affect the proliferation of the breast cancer cells. MK571 was the only drug to affect the cell migration. Ceefourin 1 and 2 and Indomethacin, which inhibit ABCC4, had no effects. This might suggest that ABCC1 plays a role in the proliferation of breast cancer cells. Similarly, it might suggest that ABCC1, or maybe both ABCC1 and ABCC4 together, are involved in the migration. However, one of the challenges associated with using inhibitors of multidrug transporters is the lack of specificity. In addition to inhibiting ABCC1 and ABCC4, MK571 can inhibit other ABCC family members, and it is also a leukotriene antagonist, inhibiting the binding of leukotriene D_4_ (LTD_4_) to cysteinyl leukotriene receptor 1 [50]. Reversan is an inhibitor of ABCC1 but can also inhibit the related protein ABCB1 [54]. Indomethacin inhibits ABCC4 but was developed as an inhibitor of cyclooxygenase. Therefore, the effect of the genetic knockdown of ABCC1 and ABCC4 was investigated.

### 2.4. Knockdown of ABCC1 and ABCC4 Using siRNA

Knockdown of ABCC1 and ABCC4 was carried out using siRNA. Two different siRNA sequences for each protein were tested, alongside a negative siRNA. The effectiveness of the knockdown was measured by both RT-qPCR to monitor the mRNA levels and Western blotting to monitor the protein levels. As can be seen in Figure 5, the knockdown of either ABCC1 or ABCC4 was effective in both cell lines. In addition, since the two proteins have overlapping substrate specificity, the dual knockdown of both proteins was also undertaken (Figure 5e).

### 2.5. The Effect of ABCC1 and ABCC4 Knockdown on Breast Cancer Proliferation and Migration

Having established that the ABCC proteins could be knocked down, the effect of this on cell proliferation was investigated. In Figure 6a, it can be seen that one of the knockdowns of ABCC1 in MCF-7 cells caused a decrease in clonogenic capacity. The double knockdown of both ABCC1 and ABCC4 caused an even larger impact. For MDA-MB-231 cells, only one of the double knockdowns had a significant effect (Figure 6b). In contrast, when examining the bulk proliferation, it can be seen in Figure 6c,d that knockdown had no significant effect on the growth of either cell line.

Next, the effect of ABCC knockdown on cell migration was investigated. Figure 7a,b shows the average results from the scratch assays. With the MCF-7 cells (Figure 7a), a knockdown with ABCC4 siRNA #35 caused a significant decrease in migration, and with the MDA-MB-231 cells (Figure 7b), both ABCC4 knockdowns caused a significant decrease in migration. To examine this further, invasion rather than just migration was investigated (Figure 7c,d). With the MCF-7 cells, no significant effects were observed; however, with MDA-MB-231 cells, the knockdown with ABCC4 siRNA #35 and one of the double knockdowns did cause a significant decrease in invasion.

These results with the ABCC knockdowns correlate well with the inhibitor studies, with ABCC1 and the double knockdown having an impact on cellular proliferation, whilst ABCC4 and the double knockdown affect migration.

### 2.6. Investigation into Potential Mechanisms by Which ABCC Transporters Affect Cellular Proliferation or Migration

ABCC1 and ABCC4 both transport a wide array of different molecules with the potential to impact cellular proliferation and migration, including cyclic nucleotides, eicosanoids, and lipid mediators such as S1P and LPI. It has previously been proposed that ABCC1 is involved in creating a feedback signaling loop with the G protein-coupled receptor, GPR55, whereby ABCC1 exports LPI, which binds to GPR55 and activates it, leading to downstream signaling and increased proliferation [31]. Therefore, we measured the expression of GPR55 in the breast cancer cell lines by Western blot (Figure 8a). GPR55 is indeed expressed in the MDA-MB-231 cells; however, there was little, if any, expression in the MCF-7 cells. Since the effects we observed with ABCC1 and ABCC4 inhibitors and knockdown were with both cell lines, this would argue against it being due to LPI export and GPR55 activation.

Next, the efflux of cAMP was investigated as a potential mechanism, as ABCC4 is capable of effluxing cyclic nucleotides [32]. An ELISA was used to assay cAMP in the cell medium (Figure 8b). The concentration of cAMP in the medium from MCF-7 cells was outside the standard range and, therefore, could not be reliably measured. For MDA-MB-231, the knockdown of ABCC1 had no significant effect on the concentration of cAMP in the medium, which is unsurprising, as ABCC1 is not reported to transport cyclic nucleotides. However, the knockdown with ABCC4 siRNA #35 did significantly decrease the concentration of cAMP in the medium, as did the treatment with MK571.

An efflux of eicosanoids such as prostaglandins could be another possible way in which ABCC4 mediates an effect [33]. An ELISA was used to measure Prostaglandin E_2_ (PGE_2_) levels in the cell medium. The concentration of PGE_2_ in the medium from MDA-MB-231 cells was outside the range of standards used and, thus, could not be reliably measured. The concentration of PGE_2_ in the medium from MCF-7 cells can be seen in Figure 8c. Neither the knockdown of ABCC proteins nor the treatment with MK571 had any effect on the levels of PGE_2_ measured.

Finally, S1P was investigated, as there are reports that both ABCC1 and ABCC4 are capable of transporting S1P [29,38]. Figure 8d shows the results of the S1P ELISA. Only the MDA-MB-231 cells effluxed sufficient S1P for reliable measurement. Although none of the siRNA knockdown samples showed any significant difference in S1P concentrations, the treatment with MK571, which inhibits both ABCC1 and ABCC4, did significantly decrease the amount of S1P in the medium.

## 3. Discussion

ABCC1 and ABCC4 have previously been implicated as negative prognostic indicators for breast cancer [16,17,18,19,22]. This may be due to their ability to export chemotherapeutic drugs, leading to multidrug resistance [13,14,15,16,17,18,19,20,21]. However, ABCC1 and ABCC4 can also transport a wide range of substrates involved in inflammation and cellular signaling and have been shown in several different cancer types to have a role in cancer development and progression that is independent of drug resistance [39,40,41,42,43,44,45,46,47,55]. Therefore, in this study, we investigated whether ABCC1 or ABCC4 are involved in cellular proliferation or migration in breast cancer cell lines. We found that both MCF-7 and MDA-MB-231 cells expressed both ABCC1 and ABCC4, with MDA-MB-231 cells having a higher expression of both transporters, which is in agreement with previous studies [56,57]. MDA-MB-231 cells are basal-type triple-negative and often reported as more aggressive than MCF-7 cells, so this would correlate with the expression levels observed. It has been reported that ABCC1 is frequently expressed in lymph node metastases of breast cancer patients and that MRP1 expression is more pronounced in lymph node metastases than in corresponding primary tumors [58].

Using both small molecule inhibitors and siRNA knockdown of ABCC1 and ABCC4, we investigated the impact of these transporters on the proliferation, clonogenic capacity, migration, and invasion of the breast cancer cell lines. MK571, which inhibits both ABCC1 and ABCC4, and Reversan, which inhibits ABCC1, both had significant effects on the proliferation and clonogenic capacity of MCF-7 and MDA-MB-231 cells, suggesting a role for ABCC1 or both transporters in cellular proliferation. Similarly, one of the ABCC1 knockdowns and the double ABCC1/ABCC4 knockdowns decreased the colony formation with MCF-7 cells, although only one of the double knockdowns affected colony formation in the MDA-MB-231 cells. It should be noted that MK571 not only inhibits ABCC1 and ABCC4 but, also, other members of the ABCC family, and it is a leukotriene antagonist, inhibiting the binding of LTD_4_ to the cysteinyl leukotriene receptor 1 [50], which could also impact cell signaling. However, the similar results obtained with the knockdowns would argue that the effects observed are due to ABCC1/ABCC4. Similarly, Reversan is also able to inhibit the related protein ABCB1, but given that Reversan affected both cell lines, and only the MCF-7 cells showed a significant expression of ABCB1 (Appendix A), it is unlikely this is the cause. That knockdown of ABCC1 alone did not affect the colony formation of MDA-MB-231 cells, and only one of the double knockdowns affected it, maybe because of the higher expression level of the transporters in MDA-MB-231 cells. The siRNA method only results in knockdown, not knockout, as seen in Figure 5. Although the protein levels are decreased in MDA-MB-231 cells, they were higher to start with, so it is possible that sufficient protein remains even after knockdown. Furthermore, siRNA knockdown is only transient, and it is not known if all cells were affected or just a proportion of them. We chose to use siRNA, because with complete knockdowns, the overexpression of other ABC transporters often occurs to compensate. In the future, perhaps the use of lentiviral shRNA could be investigated [42,43,46].

When investigating cellular migration, only MK571 had a significant effect, and only with the MDA-MB-231 cells, suggesting perhaps that the dual inhibition of both ABCC1 and ABCC4 was important. However, the siRNA knockdowns showed that the ABCC4 knockdown affected migration in both cell lines. Why the reportedly specific ABCC4 inhibitors, Ceefourin 1 and Ceefourin 2 [52], did not cause any effect on the cellular migration, when ABCC4 knockdown did, is not clear. The cellular invasion assays were more variable but, again, showed an effect of ABCC4 knockdown in MDA-MB-231 cells.

Our results suggest that ABCC1 is important for breast cancer proliferation, whilst ABCC4 has a greater role in cellular migration and invasion (Table 2). This contrasts with a study investigating the role of these transporters in neuroblastoma, where ABCC4 was more associated with proliferation and ABCC1 with migration [39]. Similarly, in pancreatic cancer, ABCC4 was associated with proliferation [47]. However, in epithelial ovarian cancer, ABCC4 was associated with proliferation, migration, and invasion [45]. In agreement with our results, breast cancer cells overexpressing ABCC1 showed an increase in proliferation, which could be inhibited by MK571, and overexpression of ABCC1 enhanced tumor growth in mice [59]. Additionally, mice implanted with breast cancer tumors with varying levels of ABCC4 expression showed no differences in tumor growth, but an increased ABCC4 expression was associated with increased metastases [56]. Therefore, the role(s) played by these transporters is not necessarily the same across different cancers, but within breast cancers, our results appear to agree with other studies in the literature. It should also be noted that, although the current study only investigated ABCC1 and ABCC4, they are not necessarily the only important transporters. In neuroblastoma, ABCC3 was shown to be very important, albeit as a positive prognostic indicator [39]. In contrast, ABCC3 was implicated in breast cancer stem-like features [20]. ABCC11 has also been implicated in aggressive breast cancer and prognosis [18].

Understanding the mechanism by which these transporters mediate their effects is important. Both transporters are able to efflux a wide range of different molecules. It is important to know what molecule is being effluxed that causes the effects observed. It might be that targeting the ABC transporters themselves is not the best approach, as they have historically proven difficult to target, often because they have so many important roles. Instead, understanding the pathway they are involved with and targeting something earlier or later in the signaling pathway might be a more effective approach. Therefore, preliminary investigations into how ABCC1 and/or ABCC4 might elicit their effects on breast cancer cell proliferation and migration were undertaken. ABCC1 has previously been linked to GPR55 in a feedback loop, where ABCC1 exports LPI, which binds to and activates GPR55, leading to downstream signaling, and this was implicated as important for prostate and ovarian cancer cell proliferation [31]. However, we found little/no expression of GPR55 within MCF-7 cells. Given that ABCC1 inhibition or knockdown affected the proliferation and clonogenic capacity of MCF-7 cells, this would argue against LPI export and GPR55 activation being a key feature to explain our findings. The export of cAMP was a second method explored, as ABCC4 is known to be able to efflux cyclic nucleotides [32]. The knockdown of ABCC4 or inhibition with MK571 significantly decreased the amount of cAMP effluxed from MBA-MB231 cells. This agrees well with a previous study [57], and it has been suggested that increasing cellular cAMP levels is a potential method of combatting triple-negative breast cancer [57,60]. The export of PGE_2_ has also been suggested as a potential mechanism by which ABCC4 might be important in breast cancer [61]. In our study, the levels of PGE_2_ exported by MDA-MB-231 cells were too high to quantify accurately. With MCF-7 cells, no effect of ABCC1/ABCC4 knockdown or inhibition was observed. Similarly, no effect of ABCC4 overexpression on the PGE_2_ efflux from MCF-7 cells was observed previously [56]. However, the same report suggests that the knockdown of ABCC4 in MBA-MB-231 cells does decrease the PGE_2_ efflux and that this is important for breast cancer metastasis [56]. Given that we observed the effects of ABCC4 knockdown on the migration of both MCF-7 and MDA-MB-231 cells, this might argue that it is not due simply to PGE_2_ efflux; however, further investigation with MDA-MB-231 cells is needed. Finally, we investigated the efflux of S1P, as levels of S1P have previously been implicated in breast cancer [62], and both ABCC1 and ABCC4 are capable of transporting S1P [29,38]. Although the knockdown of ABCC1 or ABCC4 did not show a significant effect on extracellular levels of S1P, the inhibition of both ABCC1 and ABCC4 with MK571 did decrease the amount of S1P effluxed. This result agrees with a previous study that also observed a decreased S1P efflux from MCF-7 cells in the presence of MK571 [63]. The same study also showed that siRNA knockdown of ABCC1 decreased the amount of S1P exported from MCF-7 cells, but a large effect was only seen when the cells were stimulated with estradiol, which we did not do, and this might explain why we see no effects of ABCC1 or ABCC4 knockdown.

## 4. Materials and Methods

### 4.1. Cell Culture

The MDA-MB-231 and MCF-7 human breast cancer cells were originally obtained from ATCC. They were cultured in Dulbecco’s modified Eagle’s medium (DMEM) (Lonza, Slough, UK) containing 4.5-g/L glucose and L-glutamine supplemented with 10% *v*/*v* fetal bovine serum (FBS) (Gibco^®^, ThermoFisher Scientific, Loughborough, UK) and 100-U/mL penicillin and 0.1-mg/mL streptomycin (sterile-filtered penicillin-streptomycin solution 100×, Biowest, Nuaillé, France). Cells were cultured in an environment of 95% air and 5% CO_2_ at 37 °C.

### 4.2. Colony Formation Assay

Cell growth from a single cell was determined using the colony formation assay, as described previously [39,64]. Briefly, 100 cells/well were seeded in 6-well plates and cultured at 37 °C for 24 h, after which cells were treated with MK571 (Merck Life Science, Gillingham, UK), indomethacin (Merck), Reversan (Tocris Bioscience, Abingdon, UK), Ceefourin 1, or Ceefourin 2 (Abcam, Cambridge, UK). Control wells were untreated cells. After 7 days of culture, the colonies formed were fixed with 4% paraformaldehyde and stained with 0.1% crystal violet. Wells were left to dry in open air at room temperature, and visible colonies of 50 cells or more were manually counted under a light box using a transparent film and a fine marker. The sizes of the colonies were measured using ImageJ 1.52 (NIH, Bethesda, MD, USA).

### 4.3. Cell Proliferation Assays

Cell viability was analyzed using a standard MTT assay. Fifteen thousand MCF-7 cells or 6000 MDA-MB-231 cells were plated in each well of a 24-well plate in a total volume of 400µL of culture medium. After 4 h of culture, cells were treated with inhibitors, as described above. The cell viability was assessed at 6, 12, 24, 48, and 72 h after treatment. Forty microliters of MTT (5 mg/mL) was added to each well and further incubated for 1 h at 37 °C. Wells were left to dry at 37 °C, and 400 µL of DMSO (dimethylsulfoxide) was added and left at 37 °C for 10 min. One hundred microliters of DMSO from each well was transferred to 96-well plates in triplicates. The absorbance values were read at 570 nm using a Multiskan™ GO microplate spectrophotometer (Thermo Fisher Scientific, Gillingham, UK).

Alternatively, at the given time points after treatment, the cells were harvested, diluted in trypan blue at a ratio of 1:4, viewed using a hemocytometer and microscope, and counted manually.

### 4.4. Cell Migration Assays

Cell migration was assessed using the Cell IQ (CM Technologies—intelligent cell analysis, Tampere, Finland) scratch assay. Cells were seeded in 24-well plates to reach 100% confluency the day of the assay. A scratch across the monolayer of the cells was carefully made using a 10-µL pipette tip, and the media was replaced with fresh prewarmed culture medium. Cells were treated with the inhibitors, as described above. Three image positions were selected from each well, and images were taken at 1-h intervals. Images were analyzed using the Cell-IQ analysis software (CM Technologies).

### 4.5. Cell Invasion Assays

The 24-well plate Transwell inserts (pore size 8 µm, translucent) (Greiner bio-one, Stonehouse, UK) were coated with 30 µL of ice-cold Matrigel (Corning, VWR, Lutterworth, UK) mixed with cold serum-free medium at a ratio of 1:3. The coated Transwell inserts were left to set at 37 °C for 2 h. Fifty thousand cells were seeded in each Transwell insert suspended in 300 µL of 0.5% *v*/*v* FBS culture medium. Eight hundred microliters of chemoattractant (10% *v*/*v* FBS cell culture medium) was added to the wells, and the Transwell inserts were placed in each of the wells that contain a chemoattractant. The Transwell inserts were incubated for 24 h at 37 °C. After 24 h, the culture medium in the Transwell inserts were discarded, and a cotton swab moistened with Dulbecco’s phosphate-buffered saline (DPBS) (Biowest) was used to, gently but putting firm pressure, rub the inside of the Transwell inserts to remove cells, and this process was repeated with a second cotton swab. The cells that moved to the lower surface of the membrane of the Transwell inserts were stained with Differential Quik stain (Modified Giemsa) (Generon, Slough, UK), following the manufacturer’s instructions. The Transwell inserts were then rinsed twice with water. A cotton swab moistened with DPBS was used to gently but putting firm pressure wipe again the inside of the Transwell inserts to rid of any cells left. The Transwell inserts were left to dry at room temperature. To determine the percentage cell invasion, the invaded cells were counted from randomly selected five fields of view for each Transwell insert under a light microscope at a magnification of ×40, and the mean number of invaded cells were obtained.

### 4.6. Gene Knockdown

Gene knockdown was performed following the INTERFERin-siRNA transfection protocol (Polyplus Transfection, Illkirch, France). Briefly, 25,000 cells were seeded in 24-well plates the day before transfection, ensuring 30–50% confluency at the time of transfection. The siRNA transfection complexes were added to cells dropwise and left to incubate for 3 days at 37 °C. The siRNA used were two ABCC1 siRNA (siRNA id: SASI_Hs02_00338730 and SASI_Hs02_00338731, referred to as #30 and #31, respectively) (Gene id: 4363) and 2 ABCC4 siRNA (siRNA id: SASI_Hs02_00324134 and SASI_Hs02_00324135, referred to as #34 and #35, respectively) (Gene id: 10257) (Merck). A negative control siRNA (Merck) was used to distinguish sequence-specific silencing from nonspecific effects.

### 4.7. Whole Cell Lysates

Harvested cells were placed on ice. Three hundred to four hundred microliters of ice-cold lysis buffer (0.15-M NaCl, 0.05-M Tris, pH 8.0, 1% (*v*/*v*) Triton-X100, 1-mM EDTA, and 1-mM pepstatin, 1.3-mM benzamidine, and 1.8-mM leupeptin) was added to the cells. The cells were resuspended by vortexing. The cell suspension obtained was kept on ice and vortexed every 10 min for 1 h. Cells suspension were then centrifuged for 15 min at 15,600× *g* at 4 °C, and the supernatant containing the whole cell lysate was quantified for total protein concentrations using a bicinchoninic acid kit (Pierce, ThermoFisher Scientific). The samples were mixed with sample buffer for loading on sodium dodecyl sulphate (SDS) gels.

### 4.8. Cell Membrane Extraction

Cell pellets harvested and resuspended in 20 mL of homogenization buffer (50-mM Tris, pH 7.4, 250-mM sucrose, and 0.25-mM CaCl_2_) containing protease inhibitors (1-mM pepstatin, 1.3-mM benzamidine, and 1.8-mM leupeptin). All the subsequent steps were done on ice. The cell suspension was then placed in the cell disruption vessel for nitrogen cavitation at 500 psi (Model 4639 Parr Cell Disruption Vessels (Parr^®^)). The vessel was incubated on ice for 15 min. Pressure was released slowly, and sample collected dropwise and centrifuged for 10 min at 560× *g* at 4 °C. The supernatant was collected, and the pellet, which contained unbroken cells and debris, was discarded. The supernatant was then ultracentrifuged at 100,000× *g* at 4 °C for 20 min. The supernatant was carefully discarded and the membrane pellet resuspended in 0.5–1-mL Tris sucrose buffer (TSB) (50-mM Tris-HCl, pH 7.4, and 0.25-mM sucrose). A needle syringe (0.45 mm × 13 mm) was used to resuspend the membrane proteins in TSB. Protein concentration was determined using a bicinchoninic acid kit (Pierce, ThermoFisher). Samples were kept on ice during the membrane preparations and were mixed with a sample buffer for loading on sodium dodecyl sulphate (SDS) gels.

### 4.9. RNA Isolation and Quantitative Real-Time (RT-qPCR)

The total RNA from MDA-MB-231 and MCF-7 cells, respectively, was extracted using a Bioline ISOLATE RNA mini kit according to the manufacturer’s protocol (Meridian Biosciences, London, UK). The quantity of the isolated RNA obtained was measured using a NanoDrop™ 1000 spectrophotometer (Thermo Fisher Scientific), and samples were stored at −80 °C. mRNA (600 ng) was reverse-transcribed to cDNA using a Precision NanoScript™ 2 Reverse Transcription kit (Primerdesign, Chandlers Ford, UK) according to the manufacturer’s protocol. The samples were then placed in a thermocycler, which was set for 20 min at 42 °C, 10 min at 75 °C, and holding at 4 °C. The cDNA samples obtained were diluted 1 in 10 with RNase/DNase-free water and stored at −20 °C.

The cDNA samples were amplified in qPCR using the Thermo Scientific PiKoReal 96 Real-Time PCR system. For one reaction, a master mix containing 10 µL of PrecisionPlus™ 2 × qPCR Mastermix with Sybr green with inert blue dye (Primerdesign), 1 µL of the reverse primer, 1 µL of the forward primer, and 3 µL of RNase/DNase-free water was prepared. Five microliters of cDNA were first added to each well of a 96-well Piko PCR plate in triplicate, followed by 15 µL of the master mix, ensuring that no bubbles were formed in the wells. The expression levels were normalized to human β-actin and human GAPDH as the house keeping genes. RNase/DNase-free water was used as the negative control. The PCR primers were used as follow: β-actin forward nucleotide: 5′-CTGGAACGGTGAAGGTGACA-3′, reverse nucleotide: 5′-AAGGGACTTCCTGTAACAATGCA-3′; GAPDH forward nucleotide: 5′-TGCACCACCAACTGCTTAGC-3′, reverse nucleotide: 5′-GGCATGGACTGTGGTCATGAG-3′; ABCC1 forward nucleotide 5′-CGACATGACCGAGGCTACATT-3′, reverse nucleotide 5′-AGCAGACGATCCACAGCAAAA-3′; and ABCC4 forward nucleotide 5′-TGTGGCTTTGAACACAGCGTA-3′, reverse nucleotide 5′-CCAGCACACTGAACGTGATAA-3′. The qPCR data were analyzed using the double-delta Ct analysis. Each reaction was performed three times.

### 4.10. Western Blotting

For Western blot analysis, the total protein samples (80 μg/well) were loaded and separated by SDS-PAGE (sodium dodecyl sulphate polyacrylamide gel electrophoresis), then transferred to a PVDF membrane. The membrane was blocked in blocking buffer (5% *w*/*v* BSA in Tris-buffered saline (25-mM Tris, pH 7.4, 150-mM NaCl, and 0.05% Tween 20) (TBS-T) for 1 h at room temperature, then incubated in appropriate primary antibodies for 1 h at room temperature or overnight at 4 °C. The primary antibodies used were: anti-ABCC1 derived in rabbit (1:1000, EPR4658(2); Abcam), anti-ABCC4 derived in rat (1:100, M_4_I-10; Abcam), anti-ABCB1 derived in mouse (1:200; Thermo Fisher), anti-ABCG2 derived in mouse (1:50, Bxp1; Santa Cruz, Heidelberg, Germany), or anti-GPR55 (rabbit) (1:100; Abcam). The secondary antibodies used were anti-rabbit HRP (1:3000; Cell Signaling Technology, Leiden, The Netherlands), anti-rat HRP (1:5000; Merck), and anti-mouse HRP (1:4000; Cell Signaling Technology). Blots were visualized using a SuperSignal West Chemiluminescent kit (Pierce, Thermo Fisher) and the Li-Cor C-Digit blot scanner, and the images were analyzed using the Image Studio Lite software imaging system (Li-Cor, Cambridge, UK). Afterwards, blots were re-probed using anti-α-tubulin (mouse) (1:1000; Merck) to check for equal sample loading. Western blot analysis was repeated at least 3 times for each experiment.

### 4.11. ELISA

Following siRNA knockdown, to determine the levels of molecules exported from cells, ELISAs were carried out on the media. As a control, 3 wells of nontransfected cells were treated with 50-µM MK571. For cAMP, cells were first stimulated with 100-µM forskolin for 2 h at 37 °C. For PGE_2_, cells were stimulated with 10-µg/mL lipopolysaccharides (LPS) for 24 h at 37 °C and with 80-nM phorbol 12-myristate 13-acetate (PMA) for 1 h at 37 °C, respectively. For S1P, there was no stimulation. The cell culture medium was then replaced with a fresh cell culture medium, and the cells were incubated for 18 h at 37 °C. The supernatants (conditioned media) were centrifuged at 1000× *g* to pellet out any floating cells and collected and stored at −80 °C or used immediately for determination of cAMP by using the cAMP ELISA kit (Enzo Life Sciences Inc., Exeter, UK), for the determination of PGE_2_, using the PGE_2_ ELISA kit (Enzo Life Sciences Inc.), or for S1P, using the human S1P ELISA kit (Abbexa Ltd., Cambridge, UK), according to the manufacturer’s instructions. The cells were also harvested for protein quantification.

### 4.12. Statistical Analysis

Statistical analysis of data was carried out using GraphPad Prism 8.1 (San Diego, CA, USA). For multiple comparisons with one independent variable, a one-way ANOVA was used with Dunnett’s post-hoc test. For multiple comparisons with two independent variables, a two-way ANOVA was used with Dunnett’s post-hoc test. A value of *p* < 0.05 was considered significant.

## 5. Conclusions

In summary, our results show that ABCC1 plays a role in breast cancer proliferation, whilst ABCC4 has a greater role in cellular migration and invasion. It may well be that both transporters are important; their overlapping substrate specificity means they can likely compensate for each other. The mechanism by which these transporters (and others) are involved in the development and progression of breast cancer needs to be investigated further. It is key to know exactly how they are involved before they can be considered as targets for treatment.

## Figures and Tables

**Figure 1 ijms-21-07664-f001:**
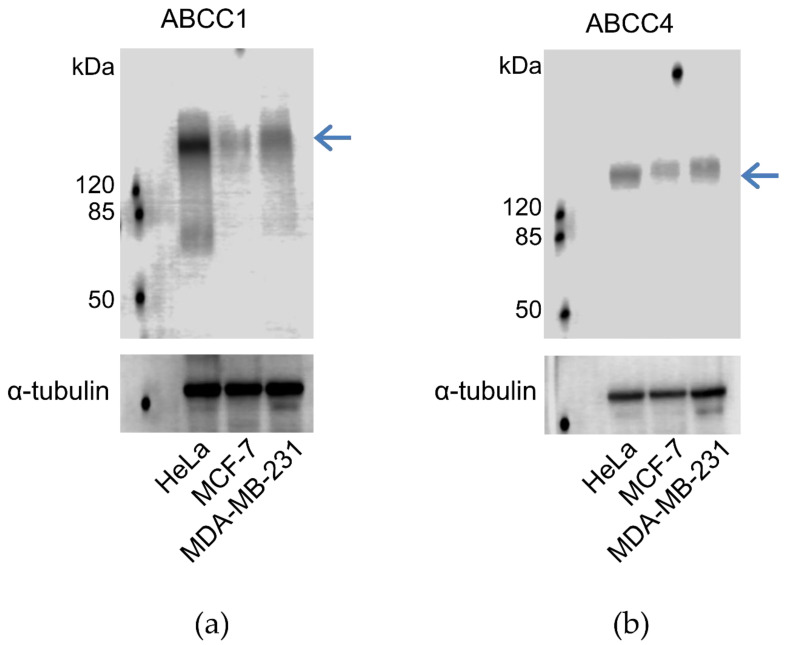
ABCC1 and ABCC4 are both expressed in MCF-7 and MDA-MB-231 breast cancer cells. Membrane extracts (80 μg protein/well) from MCF-7 and MDA-MB-231 cells were assayed by Western blot for the expression of (**a**) ABCC1 or (**b**) ABCC4, alongside Hela cells as a positive control. ABCC1 was detected using the anti-MRP1 EPR4658 antibody (1:1000). ABCC4 was detected using the anti-ABCC4 M_4_I-10 antibody (1:100). The blue arrow indicates the band corresponding to ABCC1 (**a**) or ABCC4 (**b**). Comparable sample loading was monitored afterwards using the anti-tubulin primary antibody. Uncropped images can be found in Appendix A.

**Figure 2 ijms-21-07664-f002:**
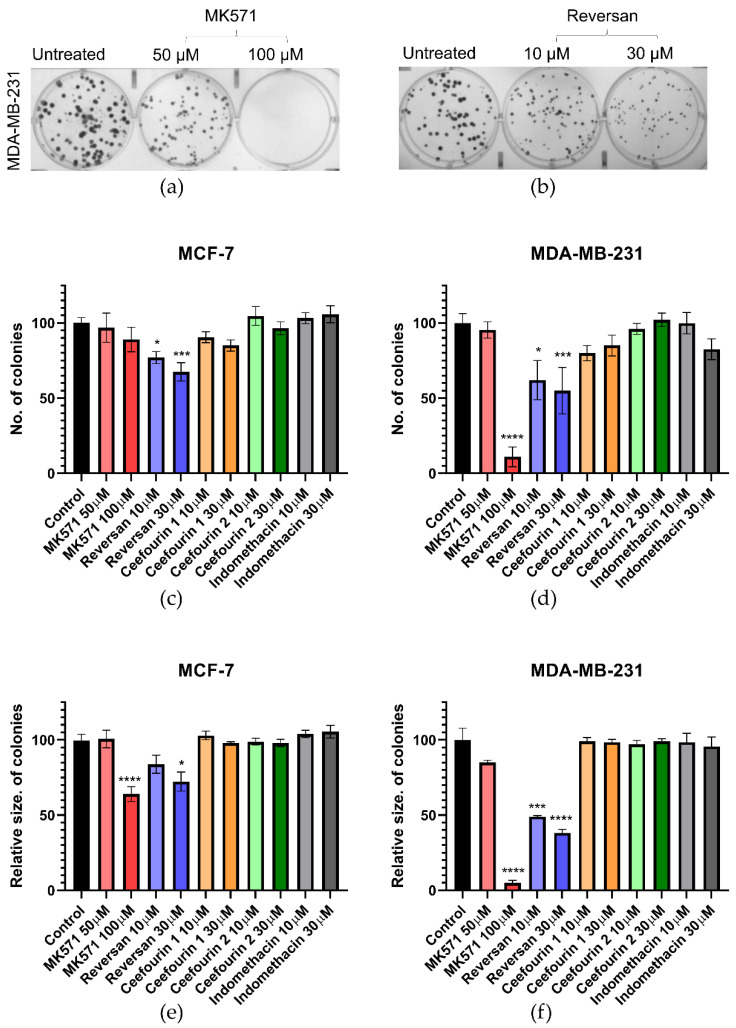
MK571 and Reversan affect the clonogenic capacity of breast cancer cells. 100 cells/well were seeded in 6-well plates and cultured at 37 °C for 24 h, after which cells were treated with the indicated concentrations of inhibitors. Control wells were untreated cells. After 7 days of culture, the colonies formed were fixed with 4% paraformaldehyde and stained with 0.1% crystal violet, the colonies counted and measured. Example results for MDA-MB-231 cells treated with (**a**) MK571 or (**b**) Reversan. Average results for the number (**c**,**d**) and size (**e**,**f**) of colonies for MCF-7 cells (**c**,**e**) and MDA-MB-231 (**d**,**f**). Data are mean ± sem, n ≥ 6. Data were analyzed using a one-way ANOVA with a Dunnett’s post hoc test; * *p* < 0.05, *** *p* < 0.001, and **** *p* < 0.0001 significantly lower than the untreated sample.

**Figure 3 ijms-21-07664-f003:**
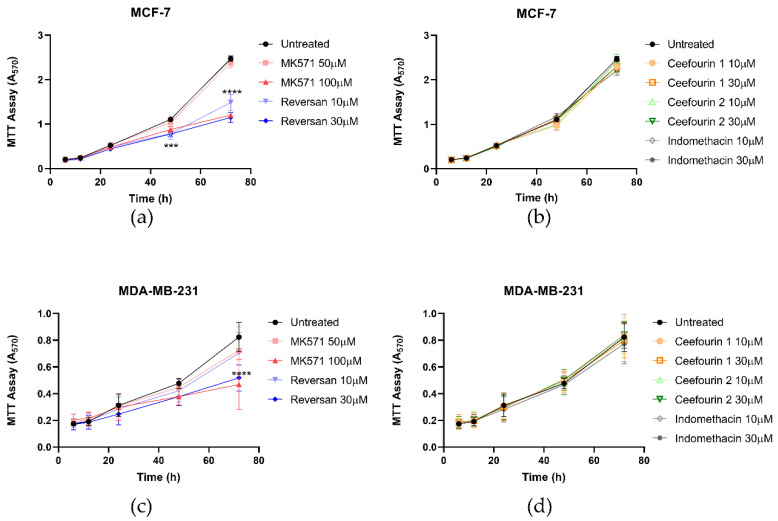
MK571 and Reversan affect the proliferation of breast cancer cells. Fifteen thousand MCF-7 cells (**a**,**b**) or 6000 MDA-MB-231 cells (**c**,**d**) were seeded in 24-well plates. After 4 h of culture, cells were treated with inhibitors, as detailed. Cell viability was assessed at 6, 12, 24, 48, and 72 h after treatment using an MTT assay and absorbance measured at 570 nm. Data are mean ± SD, n ≥ 6. Data were analyzed using a two-way ANOVA with a Dunnett’s post hoc test. *** *p* < 0.001 and **** *p* < 0.0001 significantly lower than the untreated sample.

**Figure 4 ijms-21-07664-f004:**
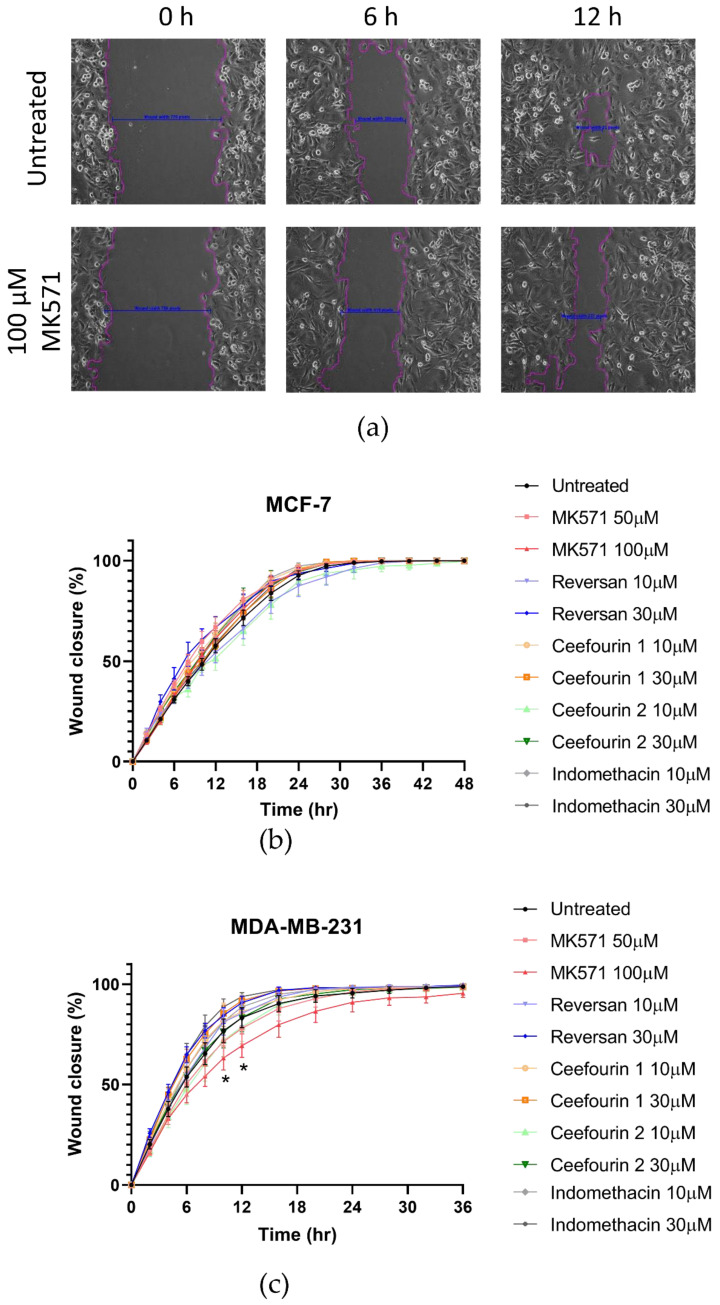
MK571 decreases the rate of migration by MDA-MB-231 cells. Cells were seeded in 24-well plates to reach 100% confluency the day of the assay. A scratch across the monolayer of the cells was carefully made, and the medium was replaced with fresh prewarmed culture medium. Cells were treated with the inhibitors as described above. Three image positions were selected from each well, and images were taken at 1-h intervals using the Cell-IQ. Representative images of MDA-MB-231 scratch assay (**a**). Pink lines represent the scratch edges as defined by the Cell IQ software, and the blue lines are the distance measurement between the edges. Average results for MCF-7 (**b**) and MDA-MB-231 (**c**) cell migration in the presence of inhibitors. Data are mean ± sem, n ≥ 6. Data were analyzed using a two-way ANOVA with a Dunnett’s post hoc test. * *p* < 0.05 significantly lower than the untreated sample.

**Figure 5 ijms-21-07664-f005:**
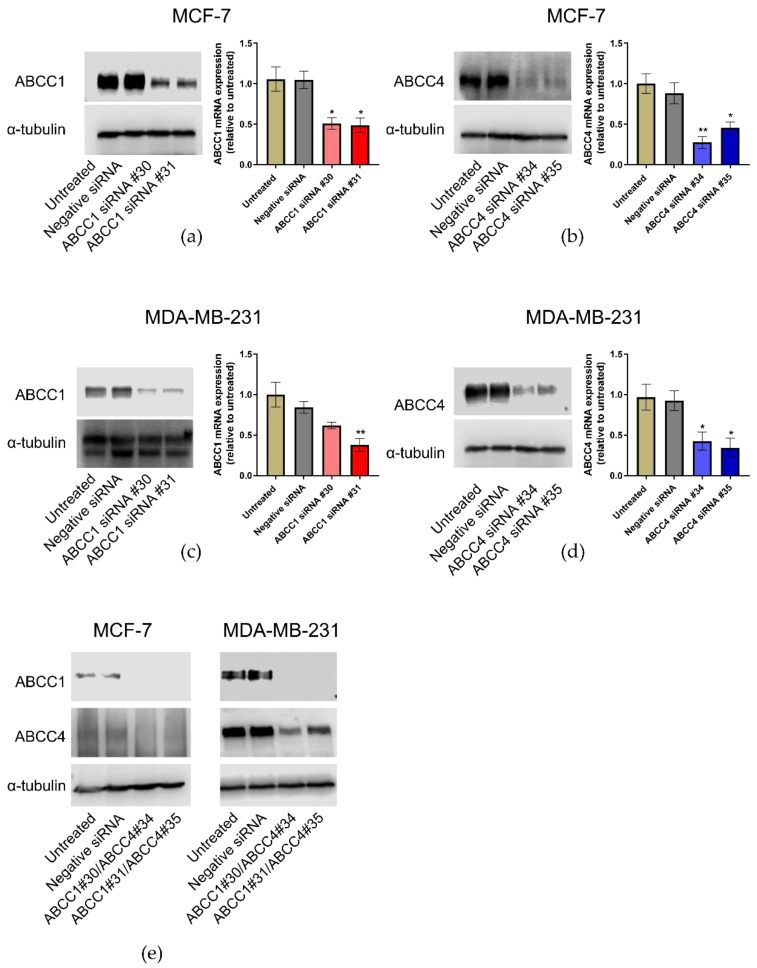
ABCC1 and ABCC4 can be successfully knocked down in breast cancer cells using siRNA. Gene knockdown in breast cancer cells was performed using the INTERFERin-siRNA transfection protocol, with two different ABCC1 siRNAs (#30 or #31), two different ABCC4 siRNAs (#34 or #35), or in combination (#30/#34 or #31/#35). A negative control siRNA was also used. The effectiveness of the knockdown was measured by both RT-qPCR and Western blotting. (**a**) Knockdown of ABCC1 in MCF-7 cells, (**b**) knockdown of ABCC4 in MCF-7 cells, (**c**) knockdown of ABCC1 in MDA-MB-231 cells, (**d**) knockdown of ABCC4 in MDA-MB231 cells and (**e**) double knockdown of both transporters in each cell line. Uncropped images can be found in Appendix A. RT-qPCR data are mean ± sem, n = 3. Data were analyzed using a one-way ANOVA with a Dunnett’s post hoc test. * *p* < 0.05 and ** *p* < 0.01 significantly lower than the negative siRNA sample.

**Figure 6 ijms-21-07664-f006:**
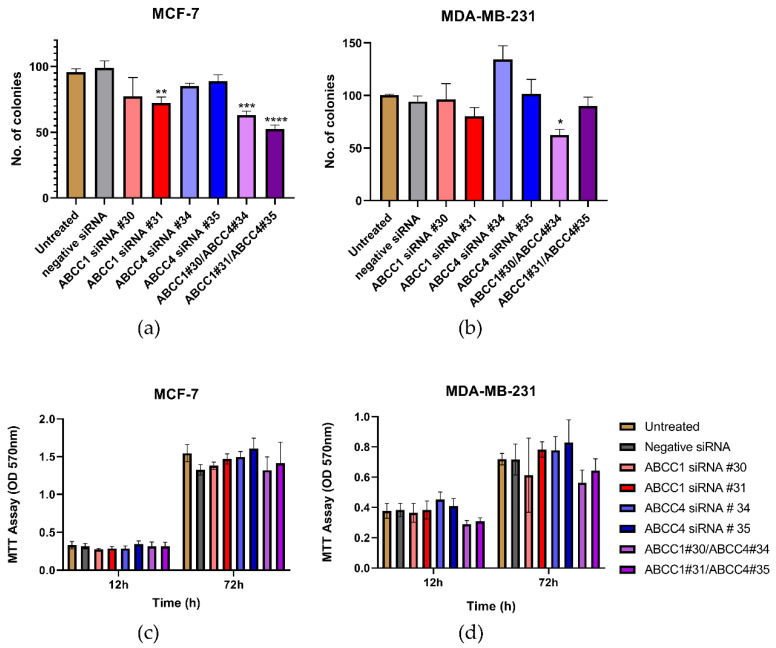
Combined knockdown of ABCC1 and ABCC4 affects the clonogenic capacity of breast cancer cells. (**a**,**b**) Clonogenic capacity of breast cancer cells following the siRNA-mediated knockdown of ABCC1 or ABCC4 was analyzed using a colony formation assay. (**c**,**d**) Proliferation of breast cancer cells following the siRNA-mediated knockdown of ABCC1 or ABCC4 was analyzed using an MTT assay. Data are mean ± sem, n ≥ 6. Data were analyzed by an ANOVA with a Dunnett’s post hoc test. * *p* < 0.05, ** *p* < 0.01, *** *p* < 0.001 and **** *p* < 0.0001significantly lower than with the negative siRNA treatment.

**Figure 7 ijms-21-07664-f007:**
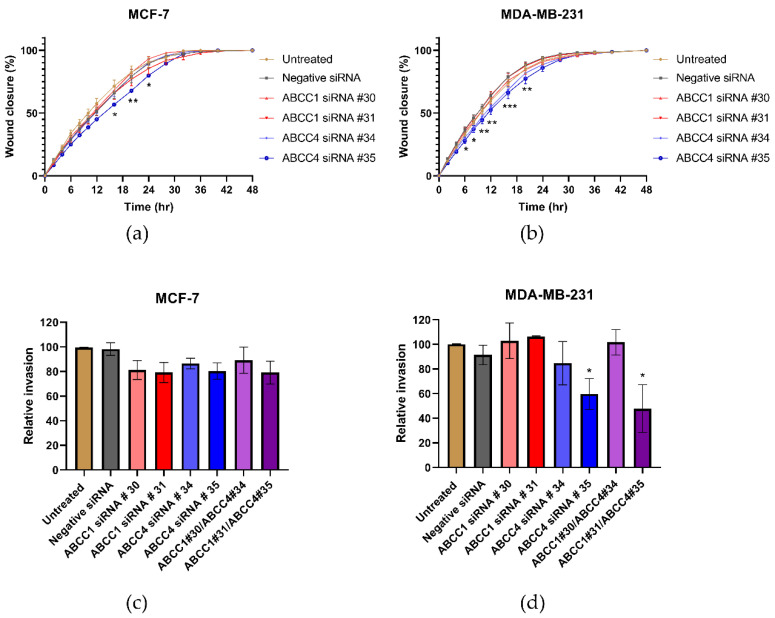
siRNA knockdown of ABCC4 affects the migration of breast cancer cells. (**a**,**b**) Migration of breast cancer cells following the siRNA-mediated knockdown of ABCC1 or ABCC4 was analyzed using a scratch assay. Data are mean ± sem, n ≥ 9. (**c**,**d**) Migration of breast cancer cells following the siRNA-mediated knockdown of ABCC1 or ABCC4 was also investigated using a cellular invasion assay. Data are mean ± sem, n ≥ 3. Data were analyzed by an ANOVA with a Dunnett’s post hoc test. * *p* < 0.05, ** *p* < 0.01, and *** *p* < 0.001 significantly lower than with the negative siRNA treatment.

**Figure 8 ijms-21-07664-f008:**
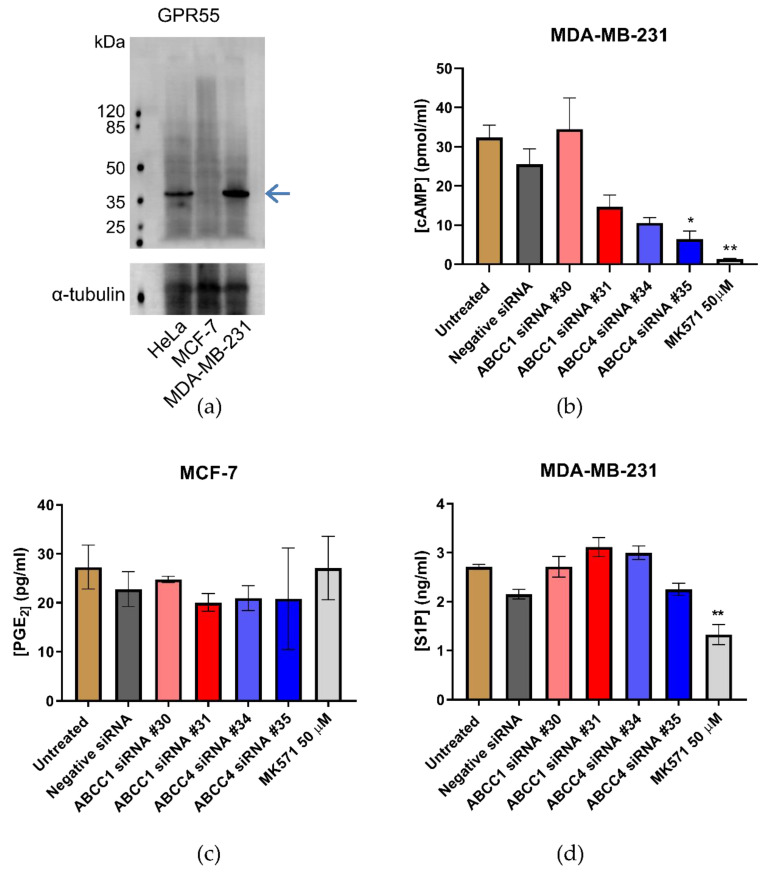
Efflux of cAMP or S1P are possible mechanisms by which ABCC transporters affect breast cancer cells. (**a**) Western blot analysis of the expression of GPR55 in breast cancer cell lines. The blue arrow indicates the band corresponding to GPR55 (**b**) Efflux of cAMP from MDA-MB-231 breast cancer cells was assayed using an ELISA. Data are mean ± sem, n ≥ 2, each in duplicate. (**c**) Efflux of Prostaglandin E_2_ (PGE_2_) from MCF-7 breast cancer cells was assayed using an ELISA. Data are mean ± sem, n ≥ 2, each in duplicate. (**d**) Efflux of S1P from MDA-MB-231 breast cancer cells was assayed using an ELISA. Data are mean ± sem, n ≥ 3, each in duplicate. Data were analyzed by a one-way ANOVA with a Dunnett’s post hoc test. * *p* < 0.05 and ** *p* < 0.01, significantly lower than untreated or negative siRNA treatment.

**Table 1 ijms-21-07664-t001:** ABCC inhibitors used in this study.

Inhibitor	Inhibits ABCC1	Inhibits ABCC4	Reference
MK571	√	√	[48,49,50]
Reversan	√		[51]
Ceefourin 1		√	[52]
Ceefourin 2		√	[52]
Indomethacin		√	[33,53]

**Table 2 ijms-21-07664-t002:** Summary of findings. ↓ significant decrease. — no significant effect. n/d not determined.

Condition	Clonogenic Capacity	Proliferation	Migration	Invasion	Comparable Findings in the Literature
ABCC1 inhibition	↓	↓	—	n/d	[59]
ABCC4 inhibition	—	—	—	n/d	
Double inhibition	↓	↓	↓	n/d	[59]
ABCC1 knockdown	↓	—	—	—	[59]
ABCC4 knockdown	—	—	↓	↓	[56]
Double knockdown	↓	—	n/d	↓

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
