# Peer review of "Roles of ABCC1 and ABCC4 in Proliferation and Migration of Breast Cancer Cell Lines"

_ijms, 2020, doi:10.3390/ijms21207664_

Round 1

Reviewer 1 Report

The current study aimed to study the roles of ABCC1 and ABCC4 in the proliferation, invasion, and migration of breast cancer cells. The results indicated that they exhibit different biological roles as ABCC1 in proliferation, whilst ABCC4 in migration and invasion. It's my opinion it can be published after minor revision.

  1. Please consider drawing a Table summarized your findings (included a comparison with former different findings);
  2. Figures (columns) can be revised to make them more attacking, e.g., keep the border color the same with column; 
  3. the results of Ceefourin 1 & 2 and Indomethacin can be moved to SI (may keep some of those important for data/mechanisms interpretation) as they showed only negative results which might work as strong noises to impact readers to get a clear view. 

Reviewer 2 Report

The topic described in the manuscript is interesting since ABC transporters are widely known to be responsible for the resistance of cancer cells to chemotherapy treatment. The authors present the analyses proving  the following function of these proteins in cancer development and progression which is not connected with the active efflux.

The analyses presented in the manuscript exhaust the goal of the study to investigate the role of ABCC1 and ABCC4 in breast carcinogenesis.

Author Response

We are very pleased that the reviewer considered our study to have achieved its aims and thank them for their comments.

Reviewer 3 Report

The manuscript of Low et al. describes the roles of ABCC1 and ABCC4 transporters in development and progression of breast cancer. Using two breast cancer cell lines, triple negative MDA-MB-231 and MCF-7 (luminal type) the authors show their roles in proliferation or migration and that another mechanisms than transport of cytotoxic drugs are involved. Using molecular inhibitors and siRNA knockdown they show that ABCC1 appears to have role in proliferation while ABCC4 has role in cellular migration (and/or invasion). They also study possible mechanisms. The results are clearly presented and convincing. Overall, the manuscript is well written and has high scientific level.

However, I have some minor comments, that should be addressed:

  • STR analysis of used cell lines should be presented.
  • Page 1 line 27. Please change “>” to „more than“.
  • Page 2 line 55. I would not use the term “recently” for the paper from 2010.
  • Page 12 lines 252-254. It is not clear how this sentence is related to the other content in the paragraph. Please explain.
  • Page 13 line 301. I agree with you that understanding underlying mechanism is important. Could you elaborate in a sentence or two - why?

Round 2

Reviewer 1 Report

Thank you for the revision. It's now publishable.